# Evaluation of the Productivity of New Spring Cereal Mixture to Optimize Cultivation under Different Soil Conditions

**Danuta Leszczyńska [1],\* , Agnieszka Klimek-Kopyra [2],\* and Krzysztof Patkowski [3]**

1    Department of Cereal Crop Production, Institute of Soil Science and Plant Cultivation—State Research Institute, Czartoryskich 8, 24-100 Puławy, Poland

2    Department of Agroecology and Plant Production, University of Agriculture in Cracow, Mickiewicza 21, 31-120 Kraków, Poland

3    Institute of Animal Breeding and Biodiversity Conservation, University of Life Sciences in Lublin, Akademicka 13, 20-950 Lublin, Poland; krzysztof.patkowski@up.lublin.pl

\*    Correspondence: leszcz@iung.pulawy.pl (D.L.); agnieszka.klimek@urk.edu.pl (A.K.-K.)

**Abstract:** The aim of the study was to evaluate grain yields, protein yields, and net metabolic energy yields of different combinations of spring types of barley, oat, and wheat arranged in 10 mixtures and grown under different soil types. Naked cultivars of barley and oat were used. The three-year field experiment was conducted at the Agricultural Advisory Centre in Szepietowo, Poland. The study showed that the major factor determining yields of the mixtures was soil quality. Within the better soil (Albic Luvisols), the highest yield was achieved by a mixture of covered barley and wheat and by a mixture of covered barley with covered oats and wheat, but only in treatments with lower sowing density. Moreover, on the better soil, significantly higher protein yields were obtained for mixtures of barley (covered or naked grains) with wheat as compared to the mixture of covered barley with covered oats, or the mixture of covered barley with naked oats and wheat. The highest yields of net metabolic energy, regardless of soil type, were obtained from a mixture of naked barley with wheat, while the lowest from a mixture of covered barley with naked oats and wheat. Mixed sowings increase biodiversity of canopies, which allows a better use of production space. They also increase health and the productivity of plants.

**Keywords:** spring cereal mixtures; grain yield; protein yield; metabolic energy yield; differentiations of cereal mixture; sustainable agriculture

## 1. Introduction

A growing demand for food in developed and developing countries as well as natural disasters such as drought, disease, and pests are becoming a major challenge for agricultural production in the 21st century. Today, purely species-specific crops dominate the cultivation of cereals. The opposite option to pure sowing of cereal species may be cereal mixtures, mainly interspecies, which are currently estimated to account for 1% of this group of agricultural crops [1]. Cereals and cereal-and-legume mixtures are an essential link in the transition of sustainable agriculture and organic farming [2]. In Central Europe, a disturbing trend is the high percentage of cereals in the sowing structure, which results in a succession of cereal crops for several years. Moreover, each simplification in the tillage system increases the disturbance of biological balance in the agricultural environment. The dominance of one cereal species, or of one cultivar within a species in a given area, promotes the development of pathogens, which causes a decrease in yields. Cereal mixtures maintain better plant health by increasing the biodiversity of the canopy [3,4].

The increase of plant productivity based on biodiversity is conditioned by more effective use of the interrelationships among plants in a mixed crop [5,6]. One of the concepts for increasing plant productivity in this cultivation system is to optimize plant species selection in mixed sowings to make complementary use of available space, water, and nutrients [7,8]. This concept was presented by Li et al. [9] who claimed that the complementary effect is associated with a better use of space by one of the components of the mixture, which has not been fully utilized by other components less tolerant to the given habitat conditions. It can be done by using plant architecture as a strategy to allow one member of a mix to capture sunlight that would otherwise be unused. According to Li et al. [9], this concept is considered a spatial complementarity, and the phenomenon of competitiveness in that concept is defined as the properties of the species that are characterized by faster development and better control of the space, which limits the development of other components in mixed sowing [10–12]. In mixed sowing, lower weed infestation, poorer pest infestation, and better resistance to lodging are observed due to the production of lower and more flexible stems [13,14]. Different species in mixtures better penetrate the soil thanks to their different root systems and enable a more efficient use of fertilization, which can be applied in smaller doses [3]. A two-species mixed sowing, consisting of species of different crop groups, is not a common practice in mechanised systems due to higher labour input, mainly during sowing and harvesting, as well as due to the instability in yields due to weather conditions [3,13,15]. An alternative agrotechnical solution is to compose mixtures or mixes within one group of plants, e.g., cereals. Cereal species mixtures can increase the intra-species diversity of the cropping system diversity by increasing genetic diversity in the canopy. Such use of intra-species diversity is well suited to mechanised systems that are designed to manage a single species, as it can provide benefits from reduced disease, weed, and insect pressure as well as improve yield level and quality per hectare [16,17]. In large farms, where the share of cereals in the farming systems often reaches 75% and where genetic uniformity lead to a biological imbalance in the fields, the use of multispecies cereal mixtures becomes a desirable solution [18].

In central Europe, there is a large variation in soil quality, which is a major problem in terms of increasing the productivity of cereal crops. To minimize the impact of soil variability on yield, mixed sowing is promoted. Hong et al. [6] have shown that, on small and large farms, the yield is determined by the sowing method. The larger the field area, the greater the overall variability of soil quality, which favours yields of mixtures compared to sole crop/pure stand, due to the dominance in the canopy of this cereal species for which the soil characteristics are appropriate. Cultivating naked cultivars of spring barley and oats in pure sowings, due to lower yields, is less economical than pure sowing of the covered forms of these cereals. It is recommended to cultivate naked cultivars of these cereals in mixtures intended as fodder on one's own farm due to better quality of grains of naked forms [19–21]. Mixed sowing increases biodiversity in the fields and contributes to the sustainability of crop production [3,7]. On large farms, the percentage of cereals in the monoculture is high, and crop simplifications often lead to an imbalance in biological diversity. Growing cereals in mixtures contributes to improving crop health. In addition, mixtures are more tolerant of unfavourable weather conditions and varied habitat conditions across the field than the tolerance of pure crop sowings of cereals. Due to lower susceptibility to climatic factors of limiting nature (shortage of precipitation, large temperature fluctuations), barley exhibits higher yield reliability than other spring cereals. The advantage of barley (as a component of the mixture) is the greatest resistance to drought among spring cereals due to a lower transpiration coefficient and high root suction power. Barley as a component of the mixture, in comparison with other cereal species, is very sensitive to soil acidification [22].

The inclusion of naked barley and naked oats into a mixture increases the protein and fat contents in the grains of the mixture, which contributes to their better forage value. It is advantageous to cultivate naked cereal cultivars in mixtures that are intended for fodder because of better quality of naked grain forms [23]. In the absence of the husk, the metabolised energy of the naked oat kernel can be comparable to or higher than that of wheat [24]. Naked oat kernels have also been shown to have a higher content of metabolised energy, lipids, linoleic acid, protein, essential amino acids, and starch

than husked oat cultivars [25,26]. These characteristics make naked oats potentially more suitable as a feed source than other cereals particularly for poultry (MacLeod et al., 2008). Oat grain has a number of nutritional benefits compared to other cereals [27]. It has a high lipid content compared to wheat and barley, which comprises principally unsaturated oleic and linoleic fatty acids as well as high concentrations of the amino acids' lysine, methionine, and cysteine [23]. For human nutrition, they are a source of soluble fibre and β-glucans, which both can have positive effects on health [23,28].

The oats are characterized by high phytosanitary properties and, thus, they can reduce the infestation of the mixture canopy by fungal diseases. A novelty of the study lies in the comparison of different spring species' composition of mixtures when taking into account new type cultivars (naked vs. covered) of barley and oats. This creates new possibilities to increase the yield and quality of grains without increasing the expenditure on chemical means of grain production under conditions of sustainable agriculture. In the light of unpredictable environmental variation factors, the great impediment is choosing the right cultivar or cultivar mixture. Ločmele et al. [29] highlighted that it is unclear how many cultivars and which type of cultivar should be used to compose the mixture.

The aim of the study was to compare the yield, protein yield, and net metabolic energy of different variants of spring cereal mixtures with the share of naked cultivars of spring barley and oats at different sowing densities, depending on soil quality. The scientific hypothesis assumed that grain yield differentiation within cereal mixture variants would be different than protein and metabolic energy yield diversification due to the lower grain yield, but a higher content of protein and metabolic energy in the grains of naked forms of barley and oats as well as wheat. A higher grain yield is expected from a mixture of hulled forms of spring barley and oats. However, the yield of protein and metabolic energy of mixtures with the share of naked forms of barley, oats, and wheat, should be similar to that of covered forms.

## 2. Materials and Methods

The field experiment with different combinations of spring cereal mixtures was conducted as part of field experimentation of Podlaskie Agricultural Advisory Centre in Szepietowo (AAC), Poland (52°52', 22°32'), in the years 2013–2015. Two, two-factorial field experiments (with the same treatments) on different types of soils were performed. The experiments were conducted on better-quality soil: Albic Luvisols (developed in loamy sand on loam), and on poorer quality soil: Haplic Arenosols (developed in loamy sand on sand), (Table 1). The first (random) factor was study years while the second factor was 10 sowing combinations. Mixture variants differed in the species composition—hulled covered barley (*Hordeum vulgare* L. cv. Skarb), naked barley (cv. Gawrosz), covered oats (*Avena sativa* L. cv. Krezus), naked oats (*Avena nuda* L. cv. Nagus), and wheat (*Triticum aestivum* L. cv. Nawra), as well as in sowing density (Table 2). The field experiment was carried out in four replications and the size of a single plot was 15 m$^2$ (length 10 m, width 1.5 m). Each plot consisted of 12 rows with a row-spacing of 12.5 cm. Grains were treated with thiram (37.5%) and carboxin (37.5%) (Oxafun T) and sowed using an Oyjord plot drill to a depth of 4 cm. After that, the sowing plots were harrowed, using a light harrow. The cultivation of barley, wheat, and oats in pure sowing was well recognized in earlier studies by the authors [22,30]. Leszczyńska and Noworolnik [22] proved that productivity of covered oat and barley is related to soil quality. In rich soil (clay soil), oat yields at 4.9 t ha$^{-1}$ and barley on 5.36 t ha$^{-1}$, while in pure (sandy soil) oat yields at 4.46 t ha$^{-1}$ and barley on 3.87 t ha$^{-1}$. Szmigiel and Oleksy [30] indicated that cultivation of covered cultivars of oat or barley in pure sowing was more beneficial than cultivation of naked cultivars of this species. The covered oat cv. 'Chwat' yielded 58% higher (5.85 t ha$^{-1}$) compared to naked cv. 'Akt' (3.71 t ha$^{-1}$) while covered barley cv. 'Rodos' yielded 11% higher (4.41 t ha$^{-1}$) compared to naked cv. 'Rastik' (3.97 t ha$^{-1}$). Zając et al. [18] indicated that wheat in pure sowing was yielded at 8 t ha$^{-1}$.

**Table 1.** Nutrient content and soil pH of the experimental field in AAC Szepietowo in 2013–2015.

| Specification | Soil Characteristics According to New Soil Classification | | | | | |
|---|---|---|---|---|---|---|
| Soil Type | Albic Luvisols | | | Haplic Arenosol | | |
| Year | 2013 | 2014 | 2015 | 2013 | 2014 | 2015 |
| pH in KCl | 5.9 | 5.9 | 6.8 | 4.9 | 5.1 | 4.9 |
| $P_2O_5$ mg/100 g soil | 15.4 | 8.7 | 20.7 | 11.0 | 9.0 | 13.4 |
| $K_2O$ mg/100 g soil | 13.5 | 11.7 | 17.0 | 13.9 | 14.4 | 10.5 |
| Mg mg/100 g soil | 7.8 | 7.8 | 5.7 | 3.4 | 4.8 | 4.0 |

**Table 2.** Experimental scheme with spring cereal mixtures.

| Treatment | Mixture Composition | * Sowing Rate of Cereals Seed Number Per 1 $m^{-2}$ |
|---|---|---|
| I | Covered barley + covered oats (CB + CO) | 160 + 300 |
| II | Covered barley + wheat (CB + W) | 160 + 290 |
| III | Covered barley + covered oats + wheat (CB + CO + W) | 107 + 200 + 193 |
| IV | Naked barley + wheat (NB + W) | 160 + 290 |
| V | Covered barley + naked oats + wheat (CB + NO + W) | 107 + 220 + 193 |
| VI | Covered barley + covered oats (CB + CO) | 136 + 255 |
| VII | Covered barley + wheat (CB + W) | 136 + 246 |
| VIII | Covered barley + covered oats + wheat (CB + CO + W) | 91 + 170 + 164 |
| IX | Naked barley + wheat (NB + W) | 136 + 246 |
| X | Covered barley + naked oats + wheat (CB + NO + W) | 91 + 187 + 164 |

* The sowing of each component in the mixture results from the recommended quantity for each species in pure sowing in accordance with the agricultural practice.

The tillage included pre-winter ploughing. In spring, a combined implement for soil tillage was used. The seeds before sowing were treated with Scenic 080 FS (100 mL + 500 L water/100 kg grain). The row spacing was 12 cm. In the autumn, phosphorus and potassium fertilization at a dose of 20 kg $ha^{-1}$ $P_2O_5$ (triple superphosphate) and 80 kg $ha^{-1}$ $K_2O$ (potassium salt 60%) were applied. Nitrogen fertilization in the dose of 60 kg $ha^{-1}$ N (ammonium nitrate 34%) was applied before sowing. Sowing was performed between 5–15 April. At harvest, the grain yield of mixtures from each plot was weighed, and grain samples were taken to determine the yield sharing of individual partners in the mixture, 1000 grain weight, and total protein content. The harvest of cereal mixtures was carried out at the stage of full maturity of cereals (Biologische Bundesanstalt, Bundessortenamt i Chemical industry-BBCH 89) in the period from 6 to 15 August.

Herbicides were used during the years of research, including: 1. Puma Uniwersal 069 EW (content of the active substance: phenoxaprop-P-ethyl- 69 g $L^{-1}$ ethylester of 2-(4-(6-chloro-1,3-benzoxazole-2-yloxy) phenoxy)propanoic acid. 2. Secateurs 125 OD (iodosulfuron-methyl-sodium 25 g $L^{-1}$, amidosulfuron 100 g $L^{-1}$), or 3. Weedlock Trio 540 SL (mecoprop (compound of the phenoxy acid group—as potassium salt)–300 g $L^{-1}$ (24.31%) M C PA (compound of the phenoxy acid group—as potassium salt)—200 g $L^{-1}$ (16.20%) dicamba (a compound from a group of benzoic acid derivatives—in the form of potassium salt) 40 g $L^{-1}$ (3.24%), which effectively destroyed dicotyledonous weeds. The problem (in a few treatments) was the occurrence of wild oats on mixed plots with oats (these weeds were removed manually).

The fungicide Soligor 425 EC (active ingredient: prothioconazole 53 g $L^{-1}$ (5.4%), spiroxamine 224 g $L^{-1}$ (22.9%), and tebuconazole 148 g $L^{-1}$ (15.1%), were used during the growing season. In the years of the study, there was a low level of cereal leaf beetle infestation of cereals below the economic harmfulness threshold.

The grain yield was determined at 15% humidity. Protein content was determined by the Kiejdahl method in the Main Laboratory of Chemical Analyses of IUNG-PIB in Pulawy. The grain energy value of cereal mixture was determined (taking into account the share of components) by converting the grain yield into net energy (MJ), when assuming the values calculated for pigs based on animal nutrition standards [31].

The yield suppression ratio (YSR) of the individual components of the mixture was calculated according to the methodology that Weigelt and Jolliffe presented [32]. The values of the yield suppression ratio were calculated from the ratio of the percentage share (weight) of grains of individual species in the yield to their percentage in the sowing material.

Analysis of variance (ANOVA) for randomized complete block design was performed for most data using the Statistica® software computer program package. Treatment means were compared using Tukey test at $p = 0.05$. Subsequently, six orthogonal contrast for selected treatment were performed using Statistica® software.

The following data were used to calculate the protein yield and energy value of the mixture grain yield: mixture grain yield, percentage share of components in grain yield, protein content in grain of individual components, and the value of metabolic energy of 1 kg of grain.

Methods of analyses were conducted according to methodology. Analysis of *p*-available was conducted based on the colorimetric assay and the Egner-Riehm DL method (PN-R-04023, 1996). Analysis of K-availability was conducted based on the photometric method (PN-R-04022, 1996).

## 3. Results

Meteorological conditions during the growing season of spring cereals (2013–2015) were not very diverse (Table 3). Average air temperatures during these growing periods were similar. The highest amount of precipitation in the growing period occurred in 2013, while the lowest occurred in 2015. This did not affect the differences in grain yields of mixtures during the years of research. The tendency for lower yields of mixtures on the better soil in 2015 can be explained by lower rainfall in that year and more permeable granulometric composition of the soil. The lack of precipitation, especially in May and June, is the main reason for low yields of cereals in a given year. It can be assumed that the sum of precipitation from March to July within the range of 220–250 mm is sufficient to obtain a fairly high spring grain yield.

**Table 3.** Meteorological conditions in 2013–2015 compared to the long-term (1969–2005).

| Month | Total Precipitation (mm) | | | | Daily Mean Temperature (°C) | | | |
|---|---|---|---|---|---|---|---|---|
| | 2013 | 2014 | 2015 | 1969–2005 | 2013 | 2014 | 2015 | 1969–2005 |
| March | 19 | 31 | 29 | 33 | −3.4 | 2.2 | 3.7 | 0.4 |
| April | 47 | 36 | 28 | 35 | 6.4 | 8.1 | 6.3 | 6.5 |
| May | 84 | 59 | 53 | 61 | 15.4 | 12.9 | 11.5 | 12.6 |
| June | 69 | 102 | 32 | 71 | 18 | 16.3 | 15.2 | 15.7 |
| July | 57 | 32 | 81 | 87 | 18.4 | 19.8 | 16.7 | 17.1 |
| Total | 276 | 260 | 233 | 283 | - | - | - | - |
| Mean | - | - | - | - | 11 | 11.9 | 10.7 | 10.5 |

Significant differences in grain yield, protein yield, and net energy yield in grain (in MJ) were found among treatments. The percentage of cereal species in the grain yield of mixtures was uneven (Tables 4 and 5). On soils of Albic Luvisols, barley exhibited a higher share in the yield (as compared to other components), which was followed by covered oats. On the Haplic Arenosols, the highest percentage in the yield was observed for covered oats, which was followed by covered barley. On the other hand, naked oats had the smallest share in grain yield of mixtures containing it on both soils.

**Table 4.** Yield sharing (%) of each partner in a mixture on Albic Luvisols (average 2013–2015).

| Treatment | Covered Barley | Naked Barley | Covered Oats | Naked Oats | Wheat |
|---|---|---|---|---|---|
| I (CB + CO) * | 55 | - | 45 | - | - |
| II (CB + W) | 53 | - | - | - | 47 |
| III (CB + CO + W) | 36 | - | 34 | - | 30 |
| IV (NB + W) | - | 50 | - | - | 50 |
| V (CB + NO + W) | 40 | - | - | 26 | 34 |
| VI (CB + CO) | 50 | - | 50 | - | - |
| VII (CB + W) | 56 | - | - | - | 44 |
| VIII (CB + CO + W)) | 37 | - | 36 | - | 27 |
| IX (NB + W) | - | 56 | - | - | 44 |
| X (CB + NO + W) | 44 | - | - | 23 | 33 |

* Covered barley + covered oats (CB + CO), Covered barley + wheat (CB + W), Covered barley + covered oats + wheat (CB + CO + W), Naked barley + wheat (NB + W), Covered barley + naked oats + wheat (CB + NO + W), Covered barley + covered oats (CB + CO), Covered barley + wheat (CB + W), Covered barley + covered oats + wheat (CB + CO + W), Naked barley + wheat (NB + W), Covered barley + naked oats + wheat (CB + NO + W).

**Table 5.** Yield sharing (%) of each partner in mixture on Haplic Arenosols (average 2013–2015).

| Treatment | Covered Barley | Naked Barley | Covered Oats | Naked Oats | Wheat |
|---|---|---|---|---|---|
| I (CB + CO) * | 48 | - | 52 | - | - |
| II (CB + W) | 55 | - | - | - | 45 |
| III (CB + CO + W) | 34 | - | 39 | - | 27 |
| IV (NB + W) | - | 47 | - | - | 53 |
| V (CB + NO + W) | 36 | - | - | 29 | 35 |
| VI (CB + CO) | 46 | - | 54 | - | - |
| VII (CB + W) | 58 | - | - | - | 42 |
| VIII (CB + CO + W) | 32 | - | 44 | - | 24 |
| IX (NB + W) | - | 49 | - | - | 51 |
| X (CB + NO + W) | 40 | - | - | 28 | 32 |

* Covered barley + covered oats (CB + CO), Covered barley + wheat (CB + W), Covered barley + covered oats + wheat (CB + CO + W), Naked barley + wheat (NB + W), Covered barley + naked oats + wheat (CB + NO + W), Covered barley + covered oats (CB + CO), Covered barley + wheat (CB + W), Covered barley + covered oats + wheat (CB + CO + W), Naked barley + wheat (NB + W), Covered barley + naked oats + wheat (CB + NO + W).

The grain yield of the studied mixtures was much higher on Albic Luvisols than on Haplic Arenosols, which was conditioned by weather conditions in studied years (Tables 6–8). On both these soils, there was a large variability in grain yield between mixture variants. Regardless of the soil quality, the highest grain yields were achieved with a mixture of hulled barley and covered oats at both sowing densities. On the better soil, similarly to it, the mixture of covered barley and wheat was yielded, but only at lower sowing density. On the poorer soil, higher yields were achieved by a mixture of barley with oats and wheat regardless of sowing density. On both soils, the lowest yields were recorded for mixtures of naked grain barley with wheat and of covered barley with naked oats and wheat. All types of mixtures differing in grain species' composition were yielded similarly under both sowing densities (insignificant differences between densities) (Tables 6 and 8).

**Table 6.** Yields of various spring cereal mixtures on Albic Luvisols.

| Treatment | Grain Yield t ha$^{-1}$ | Protein Yield kg ha$^{-1}$ | Net Energy Yield MJ |
|---|---|---|---|
| **Yield (Y)** | | | |
| 2013 | 5.62 c | 693 c | 50.8 c |
| 2014 | 5.89 b | 726 b | 53.2 b |
| 2015 | 5.96 a | 735 a | 53.8 a |
| *p* value | <0.001 | <0.001 | <0.001 |
| **Cereal mixtures (M)** | | | |
| I (CB + CO) ** | 6.30 a * | 710 c | 51.2 c |
| II (CB + W) | 5.97 b | 750 ab | 54.6 b |
| III (CB + CO + W) | 5.96 b | 730 bc | 51.2 c |
| IV (NB + W) | 5.70 c | 752 ab | 57.1 a |
| V (CB + NO + W) | 5.27 d | 681 d | 50.8 c |
| VI (CB + CO) | 6.00 ab | 678 d | 48.4 d |
| VII (CB + W) | 6.12 ab | 760 a | 55.7 ab |
| VIII (CB + CO + W) | 5.99 b | 724 b | 51.2 c |
| IX (NB + W) | 5.82 bc | 768 a | 56.8 a |
| X (CB + NO + W) | 5.14 d | 628 e | 49.0 d |
| *p* value | <0.001 | <0.001 | <0.001 |
| Y × M | <0.001 | <0.001 | <0.001 |

* Means not followed by the same letter are significantly different. ** Covered barley + covered oats (CB + CO), Covered barley + wheat (CB + W), Covered barley + covered oats + wheat (CB + CO + W), Naked barley + wheat (NB + W), Covered barley + naked oats + wheat (CB + NO + W), Covered barley + covered oats (CB + CO), Covered barley + wheat (CB + W), Covered barley + covered oats + wheat (CB + CO + W), Naked barley + wheat (NB + W), Covered barley + naked oats + wheat (CB + NO + W).

Comparison of means for grain yield, protein yield, and net energy yield by orthogonal contrasts depending on planting density proved that productivity of mixtures with covered barley depends on the sowing ratio and soil quality (Tables 7 and 9). In three mixture components, the difference of the sowing ratio of covered barley did not affect the grain yield, or yield quality. However, comparing the yields among two-component versus three-component mixtures indicates that covered barley significantly increases the productivity in higher crop density.

**Table 7.** Orthogonal contrast of selected treatments with covered barley depending on rate density Albic Luvisols.

| Orthogonal Contrast for Tested Mixture Combinations | *p*-Value of Lineal Orthogonal Contrast | | |
|---|---|---|---|
| | Seed Yield | Protein Yield | Net Energy Yield |
| I versus VI * | 0.001 * | 0.001 | 0.001 |
| II versus VII | 0.001 | 0.043 | 0.002 |
| III versus VIII | n.s. | n.s. | ns |
| V versus X | 0.001 | 0.001 | 0.001 |
| IV versus IX | 0.003 | 0.001 | ns |
| III, V, VIII, X versus I, II, VI, VII | 0.001 | 0.001 | 0.001 |

* n.s.- not significant.

**Table 8.** Yield of various spring cereal mixtures on Haplic Arenosols.

| Treatment | Grain Yield | Protein Yield | Net Energy Yield MJ |
|---|---|---|---|
| | t ha$^{-1}$ | kg ha$^{-1}$ | |
| **Yield (Y)** | | | |
| 2013 | 4.67 a | 589 a | 41.77 a |
| 2014 | 4.70 a | 593 a | 42.05 a |
| 2015 | 4.46 b | 563 b | 39.92 b |
| *p* value | <0.001 | <0.001 | <0.001 |
| **Cereal mixtures (M)** | | | |
| I (CB + CO) *** | 5.08 a * | 584 ab | 40.9 ab |
| II (CB + W) | 4.65 bc | 601 a | 42.4 ab |
| III (CB + CO + W) | 4.89 ab | 599 a | 41.6 ab |
| IV (NB + W) | 4.41 cd | 596 a | 43.0 a |
| V (CB + NO + W) | 4.14 d | 550 b | 41.0 ab |
| VI (CB + CO) | 5.09 a | 585 ab | 40.8 ab |
| VII (CB + W) | 4.53 c | 582 ab | 41.1 ab |
| VIII (CB + CO + W) | 4.81 ab | 587 a | 40.4 b |
| IX (NB + W) | 4.18 d | 567 ab | 40.8 ab |
| X (CB + NO + W) | 4.36 cd | 573 ab | 40.6 b |
| *p* value | <0.001 | <0.001 | <0.001 |
| Y × M | n.s. ** | n.s. | n.s. |

* Means not followed by the same letter are significantly different. ** n.s.- not significant, *** Covered barley + covered oats (CB + CO), Covered barley + wheat (CB + W), Covered barley + covered oats + wheat (CB + CO + W), Naked barley + wheat (NB + W), Covered barley + naked oats + wheat (CB + NO + W), Covered barley + covered oats (CB + CO), Covered barley + wheat (CB + W), Covered barley + covered oats + wheat (CB + CO + W), Naked barley + wheat (NB + W), Covered barley + naked oats + wheat (CB + NO + W).

**Table 9.** Orthogonal contrast of selected treatments with covered barley depending on rate density Haplic Arenosole.

| Orthogonal Contrast for Tested Mixture Combinations | *p*-Value of Linear Orthogonal Contrast | | |
|---|---|---|---|
| | Seed Yield | Protein Yield | Net Energy Yield |
| I versus VI | n.s. | n.s. | n.s. |
| II versus VII | 0.049 | 0.030 | 0.035 |
| III versus VIII | n.s. | 0.162 | n.s. |
| V versus X | 0.026 | 0.011 | n.s. |
| IV versus IX | 0.001 | 0.001 | 0.001 |
| III, V, VIII, X versus I, II, VI, VII | 0.001 | 0.017 | n.s. |

n.s.- not significant.

Cereal species differed in terms of grain protein content covered by barley (11.4–11.9% d.m.), naked barley (12.4–12.7% d.m.), covered oats (11.1–11.5% d.m.), naked oats (13.3–13.7% d.m.), and wheat (13.7–14.4 d.m.). On poorer soil, a slightly higher protein content in grain (by 0.2–0.3% d.m.) was obtained. These data are not shown for individual components of each mixture.

On the better soil, the highest protein yields in grains were produced by mixtures of barley (covered or naked) with wheat, regardless of sowing density. Low protein yields were found in mixtures of covered barley with naked oats and wheat (especially under lower sowing density) and covered barley with covered oats under lower sowing density. On the poorer soil, higher protein yields

of mixtures of barley (covered or naked) with wheat and mixtures of covered barley with covered oats and wheat were found, but only at higher sowing density (Tables 6 and 8).

The highest yield of net metabolic energy on the compared soils was recorded for a mixture of naked barley with wheat under both sowing densities as well as for a mixture of covered barley with wheat under lower sowing density. On Haplic Aerosols, the mixture of naked barley with wheat under higher sowing density gave the highest yield of net metabolic energy, while both 3-component mixtures, under lower sowing density—the lowest (Tables 6 and 8).

The yield suppression ratio of individual components of mixtures were varied (Tables 10 and 11) depending on cereal species and soil quality. The highest yield suppression ratio in the mixtures was found for covered barley, which was followed by naked barley, covered oats, and naked oats. Wheat turned out to be the least competitive in the mixture stand. On the Haplic Aerosols, covered barley and naked barley were more competitive in the mixtures (Tables 6 and 8). Oats (covered and naked) responded in the opposite way as it was more competitive on the better soil (Albic Luvisols) than on the poorer soil (Haplic Arenosols). The competitiveness of wheat was similar on both soil types.

**Table 10.** Yield suppression ratio of the mixture components on the soil of Albic Luvisols.

| Treatment | Covered Barley | Naked Barley | Covered Oats | Naked Oats | Wheat |
|---|---|---|---|---|---|
| I (CB + CO) * | 1.14 | - | 0.90 | - | - |
| II (CB + W) | 1.45 | - | - | - | 0.73 |
| III (CB + CO + W) | 1.36 | - | 1.11 | - | 0.68 |
| IV (NB + W) | - | 1.27 | - | - | 0.84 |
| V (CB + NO + W) | 1.33 | - | - | 0.94 | 0.83 |
| VI (CB + CO) | 1.15 | - | 0.89 | - | - |
| VII (CB + W) | 1.56 | - | - | - | 0.67 |
| VIII (CB + CO + W) | 1.28 | - | 1.26 | - | 0.60 |
| IX (NB + W) | - | 1.32 | - | - | 0.81 |
| X (CB + NO + W) | 1.48 | - | - | 0.90 | 0.76 |

* Covered barley + covered oats (CB + CO), Covered barley + wheat (CB + W), Covered barley + covered oats + wheat (CB + CO + W), Naked barley + wheat (NB + W), Covered barley + naked oats + wheat (CB + NO + W), Covered barley + covered oats (CB + CO), Covered barley + wheat (CB + W), Covered barley + covered oats + wheat (CB + CO + W), Naked barley + wheat (NB + W), Covered barley + naked oats + wheat (CB + NO + W).

**Table 11.** Yield suppression ratio of the mixture components on the soil of Haplic Arenosols.

| Treatment | Covered Barley | Naked Barley | Covered Oats | Naked Oats | Wheat |
|---|---|---|---|---|---|
| I (CB + CO) * | 1.31 | - | 0.78 | - | - |
| II (CB + W) | 1.39 | - | - | - | 0.73 |
| III (CB + CO + W) | 1.44 | - | 0.97 | - | 0.75 |
| IV (NB + W) | - | 1.35 | - | - | 0.79 |
| V (CB + NO + W) | 1.48 | - | - | 0.84 | 0.81 |
| VI (CB + CO) | 1.25 | - | 0.83 | - | - |
| VII (CB + W) | 1.51 | - | - | - | 0.70 |
| VIII (CB + CO + W) | 1.48 | - | 1.03 | - | 0.68 |
| IX (NB + W) | - | 1.51 | - | - | 0.70 |
| X (CB + NO + W) | 1.63 | - | - | 0.74 | 0.79 |

* Covered barley + covered oats (CB + CO), Covered barley + wheat (CB + W), Covered barley + covered oats + wheat (CB + CO + W), Naked barley + wheat (NB + W), Covered barley + naked oats + wheat (CB + NO + W), Covered barley + covered oats (CB + CO), Covered barley + wheat (CB + W), Covered barley + covered oats + wheat (CB + CO + W), Naked barley + wheat (NB + W), Covered barley + naked oats + wheat (CB + NO + W).

## 4. Discussion

The grain yield of the covered barley in three-component mixtures was much higher on Albic Luvisols soil than on Haplic Arenosols soil, which undermines the claim that multispecies mixtures, as an effect of specific biodiversity, grown on poorer soils, are capable of high yields. The results of the research proved that yields are determined by many interacting factors. Existing data show that equal proportion three-species mixtures may perform worse than those having a higher initial percentage of the species that is the most productive in a pure stand [3,33,34]. In our research, despite the application of half the shares of the individual components of the mixture in each combination in order to exclude the effect of species domination, we obtained the lowest yields of three-species mixtures. The yield of two-species mixtures of barley and oats was significantly better, but the yield level depended on the soil type. This is confirmed by earlier studies by Noworolnik and Terelak [35] who showed significantly higher yields of a mixture of barley and covered oats on an Albic Luvisols than on a Haplic Arenosols soil, which indicates a significant effect of habitat conditions on plant yields in the mixtures. Another aspect of the evaluation of oat-barley mixtures is the varietal selection conditioned by the structure of grain (covered vs. naked grain). Szumiło and Rachoń [36] demonstrated that higher grain yields can be obtained from barley mixtures (covered or naked) with hulled oats as compared to barley mixtures with naked oats. The above results indicate that the yields of a mixture is determined by the yielding biology of particular mixture components. This was proven by a study by Rudnicki and Wasilewska [37] who did not obtain significantly differentiated grain yields of mixtures of hulled barley with covered oats, covered barley with wheat, and barley with oats and wheat. Tobiasz-Salach et al. [38], in experiments with mixtures of covered or naked oats with other spring cereals, recorded significantly lower grain yields of hulled or naked grain mixtures of oats (hulled or naked grain) with wheat compared to the mixture of oats with covered or naked grain barley. Buczek et al. [39] showed that spring cereal mixtures (oats, barley, and wheat) yield at the level of 4.23 t ha$^{-1}$. On the other hand, Klima and Łabza [40] found that oats grown in mixed sowing with barley yield significantly higher than in pure sowing. In our experiment, the yield of the mixture of oat and barley sown in large sowing amounts to 6.3 t ha$^{-1}$.

The yield suppression ratio of individual components of mixtures were varied depending on cereal species. The highest yield suppression ratio in the mixtures were found for covered barley, which was followed by naked barley, covered oats, and naked oats. Wheat turned out to be the least competitive in the mixture stand. Presented results were partly confirmed by Klimek-Kopyra et al. [9]. The authors revealed asymmetric interspecific competition between species in two and three component mixtures. Wheat, despite having a high share in the mixture, did not display high productivity. Leszczyńska and Grabiński [41] and Czaban et al. [42] claimed that the interaction of plants in the canopy cannot be fully explained without the knowledge of allelopathy.

An important aspect that determines the suitability of plants for mixed sowings is the quality of the obtained grains. For the grain to be useful for industrial purposes, it has to exhibit a high protein content including at least 11.5% of protein in dry matter, and 14% of protein in dry matter, which is meant for improving the value of milling mixtures with low-quality grain [43]. The results of our research indicate that, due to the increased amount of protein in the compared mixtures, only the combinations with wheat are effective and appropriate for use in the fodder industry.

Mixtures of hulled barley with wheat grown on high quality soil were characterized by significantly higher protein (760 kg ha$^{-1}$) and metabolic energy yield (55.7 MJ). On Haplic Arenosols soil, the highest net metabolic energy yield was recorded for a mixture of naked barley with wheat at a higher sowing density, while the lowest—for both three-component mixtures at a lower sowing density. Other results were obtained by Kijora and Wróbel, [44], who proved that higher grain protein yields and net metabolic energy yields could be obtained from mixtures of covered barley with covered oats than from mixtures of barley with naked oats. Higher fat yields, however, can be obtained from mixtures of barley with naked oats.

## 5. Conclusions

The grain yield differentiations of cereal mixture variants was different than protein and metabolic energy yield due to the lower grain yield but higher content of protein and metabolic energy in the grains of naked forms of barley, oats, and wheat.

A higher grain yield was indicated from a mixture of cove forms of spring barley and oats. However, the yield of protein and metabolic energy of mixtures with the share of naked forms of barley, oats, and wheat was similar to that of covered forms.

Regardless of the soil quality and sowing density, the highest grain yields were obtained from a two-mixture component of covered barley with covered oats (Albic Luvisols: 6.3 t ha$^{-1}$ and 6.0 t ha $^{-1}$, Haplic Arenosols: 5.08 t ha$^{-1}$ and 5.09 t ha $^{-1}$, respectively). Three mixture components (CB + CO + W) lack of differentiations of cereal mixture variants in terms of yield and protein yields was noted.

**Author Contributions:** Conceptualization, D.L. and A.K.-K.; methodology, D.L.; software, A.K.-K.; validation, D.L., A.K.-K. and K.P.; formal analysis, A.K.-K.; investigation, D.L.; resources, D.L.; data curation, D.L.; writing—original draft preparation, A.K.-K., D.L., K.P.; writing—review and editing, A.K.-K., D.L., K.P.; visualization, A.K.-K.; supervision, D.L.; project administration, D.L.; funding acquisition, D.L. and K.P. All authors have read and agreed to the published version of the manuscript.

**Funding:** This research was funded by Institute of Soil Science and Plant Cultivation, State Research Institute in Puławy, Poland and was funded by University of Life Sciences in Lublin, Poland.

**Acknowledgments:** We would like to thank Kazimierz Noworolnik, who is an employee of the Department of Cereal Crop Production, the Institute of Soil Science and Plant Cultivation—the State Research Institute in Puławy, Poland for his contribution to the creation of scientific work and for long-term collaboration.

**Conflicts of Interest:** The authors declare no conflict of interest.

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
