# Peer review of "Evaluation of the Productivity of New Spring Cereal Mixture to Optimize Cultivation under Different Soil Conditions"

_agriculture, doi:10.3390/agriculture10080344_

Round 1

Reviewer 1 Report

Dear Authors
In the attached file you will find my suggestions aimed at implementing your manuscript both from a conceptual and methodological aspects in order to enhance the interesting topic you are addressing.

Best regards

-------------------------------------------------------------------------------------

Note to Author - Agriculture-869055-peer-review

Evaluation of the productivity of new spring cereal mixture combinations as an attempt to optimize cultivation under different soil conditions

Introduction

Given the extensive bibliography on topics such as intercropping (competition, complementary use of resources, agronomic and productive benefits), food and fodder quality of cereals, the introduction could have been more informative and more effective in contextualizing the research topic.

In particular, the introduction should be strongly implemented with information on the quality (food and fodder) of hasked and naked cultivars, both for barley, oats, and wheat, Some information about their adaptability to soil conditions (particularly pH) should be also added.

Some hints are given below

The term "hulled" should be replaced by "covered": most barley is what’s called “covered barley,” which means it has a tough, inedible outer hull around the barley kernel. This covering must be removed before the barley can be eaten. A less common variety, referred to as “naked” barley, has a covering, or hull, that is so loose that it usually falls off during harvesting.

Hulled (sometimes called Dehulled Barley) is covered barley that has been minimally processed to remove only the tough inedible outer hull. It is challenging to remove the hull carefully so that some of the bran is not lost – but that is what must be done for covered barley to be considered whole grain (National Barley Foods Council).

It would be useful to consult Lister, D.L., Jones, M.K. Is naked barley an eastern or a western crop? The combined evidence of archaeobotany and genetics. Veget Hist Archaeobot 22, 439–446 (2013). https://doi.org/10.1007/s00334-012-0376-9

Same replacement should be made for oats.

The majority of oats currently grown are husked oats, whereby the caryopsis or kernel is enclosed by a lignified lemma and palea, collectively termed the husk. The kernel does not thresh free of these husks during harvesting, and subsequent removal of these husks requires considerable energy input.

Oats generally tolerate acidic soils and moist climates better than either wheat or barley, making them ideally suited to Irish growing conditions. They are grown for both animal and human nutrition. However, the area of oats is restricted by limited market outlets.

Oat grain has a number of nutritional benefits compared to other cereals (Marshall et al., 2013). It has a high lipid content compared to wheat and barley, which comprises principally unsaturated oleic and linoleic fatty acids as well as high concentrations of the amino acids lysine, methionine and cysteine (Morris, 1990; Welch, 1995). For human nutrition, they are a source of soluble fibre and β-glucans, both of which can have positive effects on health (Anderson and Bridges, 1993).

The presence of these husks, which have lower digestibility than the kernel, means that husked oats have lower nutritional value than other cereals, particularly for non-ruminants that have limited ability to deal with high levels of dietary fibre (MacLeod et al.,2008).

In naked oats, which are the same species as husked oats, lignification of the lemma is much less than that in husked oats, such that at harvest, the lemma is thinner and less rigidly curved around the kernel, allowing the kernel to thresh free during harvest (Ougham et al., 1996). While all kernels of earlier cultivars did not always thresh free of the husk, more recent cultivars have been shown to completely thresh free of the husk (Valentine et al., 1997). The expression of the naked phenotype is thought to be highly heritable but can be affected by environmental conditions, particularly temperature (Lawes and Boland, 1974; Ubert et al., 2017). In the absence of the husk, the metabolisable energy of the naked oat kernel can be comparable to or higher than that of wheat (MacLeod et al.,2008). Naked oat kernels have also been shown to have a higher content of metabolisable energy, lipids, linoleic acid, protein, essential amino acids and starch than husked oat cultivars (Welch, 1995; Givens et al., 2004; Biel et al., 2009). These characteristics make naked oats potentially more suitable as a feed source than other cereals particularly for poultry (Hsun and Maurice, 1992; MacLeod et al., 2008), and some authors have reported that naked oats have high digestibility and metabolisable energy when used as a feed source for ruminants (Givens and Brunnen, 1987).

Material & Methods

-The experimental layout should be better illustrated. More appropriate terms, such as factor, treatment ecc., should be used in the description of design.

 - The experiment was repeated for three consecutive years (see tables on soil characteristics and climate trends) but the factor "year" was not considered. The Authors do not mention it in ANOVA in the presentation of the results (see tables). This is an important critical aspect of the work to be solved by the Authors

- The Authors declare "soil" as an important discriminating factor for the performance of mixtures. However, it is not considered as an experimental factor: in ANOVA neither the mean effects nor the interaction with other factors (mixture and time) are considered. Consequently, these results were not specifically presented and discussed. A complete soil characterization should be reported in order to recognise the two different levels of fertility. Taxonomic classification alone is not sufficient to qualify soil.

- The method of sowing the mixture is not reported: were the two/three species sown in the same row or in alternate rows? The degree of competition depends on the spatial distribution of the plants and this aspect is of considerable importance if the densities of the species in the mixture are the same as those of the respective pure crops.

-The "Competitiveness coefficients of the mixture components" is not appropriate to assess the effects of competition. In fact, almost all the indices used to evaluate intercropping benefits are calculated by referring to the respective pure crop (Relative yield, Land equivalent ratio, Aggressivity etc.).  The Authors did not include pure crops in the experiment and this choice does not allow to make a proper evaluation of mixtures grown in different sowing ratio. In any case, the choice of the "competitiveness index" was not well supported by adequate references. The reference 20 in the text was difficult to find and consult in the most common sources.

Results, discussion and Conclusion

Both of these sections reflect the methodological and approach failings noted above.

The tables must be provided with a more adequate description of the data presented.

Author Response

Dear Reviewer,

We are gratefull for reviewing our manuscript.

We corrected the introduction section acorring to your suggestion. We inseted references with information regarding to nutritional values of oats.

In methodology section iformation about sowing was added.

In methodology section it was mentioned that analysis of variance was conducted.  However, we did more advance analysis, and we add years as a factor. Previosuly we omited it beacuse, the aim of our study  was to compare the yield,  protein yield, and net metabolic energy of species mixtures at two levels of soil fertility using different sowing ratios.

The results section was corrected. We did two factorial analysis and we did orthogonal contarst analyis, which is dedicated for experiments which omit pure sowing.

Reviewer 2 Report

Very intersting paper, well written, but the methodology do not completely satisfy me.

The main drawback I've noticed is the lack of controls (pure crops of wheat, barley and oat) in the experiments. Maybe the results of the pure crops have been omitted to be used in another paper?

Often, reading the paper, a question rises to my mind: what about the pure crop performance with respect to the mixtures?

Then, a suggestion is to use ANOVA contrasts to analize results instead of using a multiple comparison test (Tuckey). With contrasts it is possible to test many specific hypothesis such as: are mixture with barley better then mixtures with oat? And so on.

Minor suggestions are in the enclosed file.

Author Response

Dear Reviewer,

We are gratefull for reviewing our manuscript.

In methodology section we did improvement regarding to your suggestions.

The results section was corrected. We did orthogonal contarst analyis, which is dedicated for experiments which omit pure sowing.

The all corrections are marked in manuscript.

Kind Regards

Authors

Round 2

Reviewer 1 Report

The revisions made to the manuscript are satisfactory.

Author Response

Dear Reviewer,

Thank you for acceptance of our changes. According to your suggestion we made an english correction.

Kind Regars

Authors

Reviewer 2 Report

You added the text:

"The cultivation of barley, wheat and oats in pure sowing was well recognized in earlier studies by the authors [3, 18]. 

OK, but in this contest it is very useful to report yield results of the pure crops, to make a comparison with mixtures.

Other minor corrections in the enclosed file

Author Response

Dear Reviewer,

Thank you for another comments. We did neceseery changes in manuscript. Below you will find particular changes which are visible in manuscript.

We made an english correction

L42: We corrected a sentence

L112: We reseted the previous aim of the study

L137: We added required information about species yielding in pure sowing.

L148: We corrected the forms of P and K.

L238: We removed from under the  Tables 7 and 9 statistical issues.

Kind Regars

Authors
